# Information Re-Organization Improves Reasoning in Large Language Models

**Xiaoxia Cheng, Zeqi Tan, Wei Xue, Weiming Lu**[*]
College of Computer Science and Technology
Zhejiang University
{zjucxx, zqtan, lokilanka, luwm}@zju.edu.cn

## Abstract

Improving the reasoning capabilities of large language models (LLMs) has attracted considerable interest. Recent approaches primarily focus on improving the reasoning process to yield a more precise final answer. However, in scenarios involving contextually aware reasoning, these methods neglect the importance of first identifying logical relationships from the context before proceeding with the reasoning. This oversight could lead to a superficial understanding and interaction with the context, potentially undermining the quality and reliability of the reasoning outcomes. In this paper, we propose an information re-organization (**InfoRE**) method before proceeding with the reasoning to enhance the reasoning ability of LLMs. Our re-organization method involves initially extracting logical relationships from the contextual content, such as documents or paragraphs, and subsequently pruning redundant content to minimize noise. Then, we utilize the re-organized information in the reasoning process. This enables LLMs to deeply understand the contextual content by clearly perceiving these logical relationships, while also ensuring high-quality responses by eliminating potential noise. To demonstrate the effectiveness of our approach in improving the reasoning ability, we conduct experiments using Llama2-70B, GPT-3.5, and GPT-4 on various contextually aware multi-hop reasoning tasks. Using only a zero-shot setting, our method achieves an average absolute improvement of 4% across all tasks, highlighting its potential to improve the reasoning performance of LLMs. Our source code is available at https://github.com/hustcxx/InfoRE.

## 1 Introduction

Large language models (LLMs) demonstrate powerful generative capabilities and achieve remarkable performance across a range of linguistic tasks [1–3]. However, their capabilities in performing complex reasoning tasks — an essential aspect of advanced language understanding and intelligent decision-making — still present substantial challenges [4, 5]. This has spurred researchers to explore innovative strategies [6–8] to improve the reasoning capabilities of these models.

Recently, diverse methods have been developed to enhance the reasoning ability of LLMs. For example, a notable method Chain-of-Thought (CoT) [6], incorporates a series of intermediate reasoning steps into the reasoning. CoT [6] allows for a more transparent and understandable path to the final answer, making it easier to follow the logic behind the conclusion. Building upon this foundation, subsequent approaches such as Tree of Thoughts (ToT) [7] and Graph of Thoughts (GoT) [8] are proposed to further refine the reasoning steps and enhance the accuracy and reliability of LLMs. Different from the sequential intermediate steps of CoT [6], ToT [7] and GoT [8] model the problem solving process into structured tools of tree and graph, respectively. A critical observation

---

[*]Corresponding author.

38th Conference on Neural Information Processing Systems (NeurIPS 2024).

is that these existing approaches primarily focus on improving the reasoning process of LLMs, as shown in Figure 1 (left). However, in scenarios involving contextually aware reasoning, it is equally important to first identify logical relationships of context before proceeding with the reasoning, not just improve their reasoning process. This is because logical relationships, such as parallelism, causal connections, contrasts, etc., are essential elements of reasoning [9]. Nevertheless, these existing methods often neglect this crucial step. Such an oversight can lead to a superficial understanding and interaction with the context, potentially undermining the quality of the reasoning results.

Figure 1: InfoRE (Ours) vs existing methods. In contrast to the existing methods that primarily focus on the reasoning process, our InfoRE emphasizes the re-organization of context information. The *[TEXT]* in italics indicates that it is optional in the reasoning process.

Inspired by the fact that when faced with context-aware reasoning tasks humans often first re-organize existing contextual information to uncover the logical relationships, eliminate noises, and enhance their understanding of the context, we propose an information re-organization (**InfoRE**) method, to ground reasoning by the re-organized information. As shown in Figure 1 (right), different from previous methodologies that primarily focus on refining the reasoning steps to enhance the reasoning capabilities of LLMs, our approach takes a novel direction of context re-organization. We emphasize the utilization of re-organized contextual content to explicitly present the logical relationships that are often implicit within the plain text, promoting more effective reasoning. Specifically, our re-organization method comprises two operations: extraction and pruning. The extraction first uncovers the implicit logical relationships within the contextual content by transforming the content into a MindMap structure [10]. We employ this structure because it is rich in logical relationships and encompasses multi-hop connections. Pruning is then used to further minimize noise that is irrelevant to the reasoning objective. The pruning operation uses a pre-trained BERT [11] based model trained with reinforcement learning (RL). Finally, we utilize the re-organized context to reason. This enables LLMs to deeply understand the context by clearly perceiving these logical relationships, facilitating the quality and reliability of reasoning. Besides, our information re-organization method can be integrated with existing prompt methods, like CoT [6], to further improve the reasoning ability of LLMs. To verify the efficacy of our proposed InfoRE method, we conduct experiments using Llama2-70B [2], GPT-3.5 [1], and GPT-4 [3] on various contextually aware multi-hop reasoning tasks, including claim verification [12], question answering [13], and reading comprehension [14]. Using only a zero-shot setting, our method achieves an average improvement of 4% across all tasks, highlighting its potential to improve the reasoning performance of LLMs.

Our main contributions are as follows:

- In contrast to existing methods that primarily focus on refining the reasoning steps to enhance the reasoning capabilities of LLMs, we take a novel direction of context re-organization.
- The re-organization method initially uncovers the logical relationships that encompass multi-hop connections in the contextual content by extraction, and subsequently minimizes noise by pruning.
- Experiment improvements on contextually aware multi-hop reasoning tasks across claim verification, question answering, and reading comprehension show the efficacy of our proposed method.

## 2   Related Work

**Reasoning with LLMs**    LLMs [1–3] have revolutionized the field of natural language processing (NLP) and demonstrate remarkable proficiency across a range of linguistic tasks. To further improve

the reasoning ability of LLMs has attracted considerable interest. In-context learning [1], as a promising approach to enhance the reasoning abilities of LLMs, has been verified in mathematical reasoning task [6]. In addition, CoT [6] incorporates a coherent series of intermediate reasoning steps to improve the reasoning ability of LLMs. Following this paradigm, ToT [7] and GoT [8] have also been proposed to focus on improving the structure of the intermediate chain. Then self-consistency [15] and plan-to-solve [15] focus on the reliability of the chain. Recently, the step-back prompting [16] is proposed, which obtains the high-level concept and first principles from instances by abstraction in the first step, then guides the reasoning of LLMs with the obtained concept and principles. For tasks that require multi-hop reasoning, current approaches [17, 18] generally decompose multi-hop problems into simpler sub-tasks problems relying on the in-context learning ability of LLMs. In contrast to existing methods, we take a novel direction of context re-organization to enhance the reasoning capabilities of LLMs for contextually aware reasoning tasks.

**Information Re-organization**  Information re-organization is a technique that leverages other structures to improve the clarity and comprehensibility of information. Moreover, it can also reveal conceptual relationships implicit in the original textual text, making it a valuable strategy for various tasks. For example, SG [19] re-organizes the document to scattered knowledge graph triples for explainable multi-hop question answering. GraphRAG [20] build a community hierarchy and generate summaries for these communities after extracting a knowledge graph, then leverage these structures on RAG-based tasks. MindMap [10] is a powerful structure method for representing knowledge and concepts, it can be used to construct a hierarchical abstraction of natural language text [21]. Previous method [22] uses it to organize a large amount of scientific material to enhance the clarity of the text material. Different from the scattered knowledge graph triples used in SG [19], MindMap is more centralized, which aggregates content related to the same topic together. In this paper, we use it as the structure of re-organized information to uncover the logical relationships and multi-hop connections implicit within the plain context with one step, as opposed to GraphRAG [20] separately aggregates information with multiple steps.

## 3 Methodology

In this section, we first give a formulation of a context-aware reasoning task in §3.1 and then describe our method in detail. As shown in Figure 2, the framework of our method consists of two components, an information re-organization §3.2 and a reasoning step using the re-organized context §3.3.

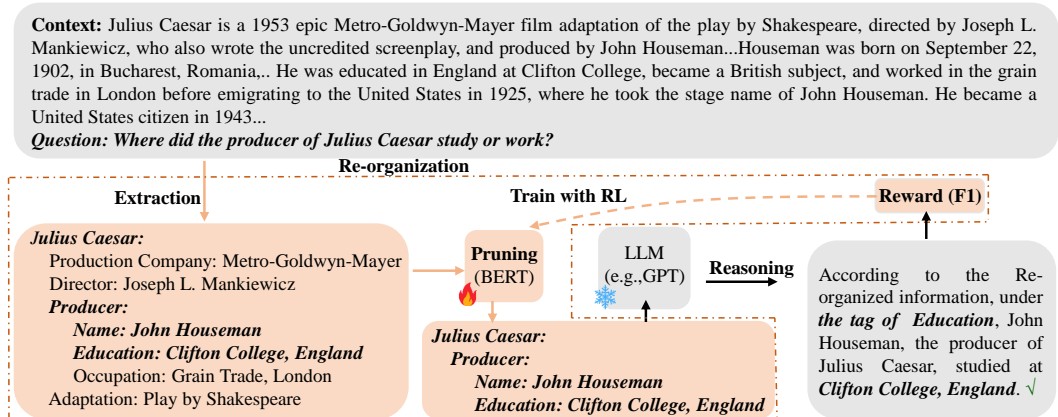

Figure 2: Illustration of our information re-organization method with two modules: 1)Information Re-Organization, which includes logic relationship extraction and noise pruning. 2) Reasoning using re-organized context. The re-organized context in ***black italicized*** text is relevant to the question.

### 3.1 Task Formulation

Given a sample $(c, q, a)$ from sample space $(\mathcal{C}, \mathcal{Q}, \mathcal{A})$, where $c$ denotes gold context, $q$ denotes question, the task aims to obtain answer $a$ to the question $q$, based on gold context $c$ with a large language model. In particular, it requires multi-hop reasoning to get the answer $a$ to question $q$.

## 3.2 Information Re-Organization

The purpose of the information re-organization is to obtain the logical relationships from the context and minimize the noise irrelevant to the question. This goal is achieved through two operations: extraction and pruning.

### 3.2.1 Extraction

For a question $q$, if the derivation of its answer $a$ relies on context $c$, then a deeper understanding of $c$ is crucial. Logical relationships, such as parallelism, causal connections, contrasts, etc., are essential elements of understanding and reasoning [9]. However, the logical relationships in the plain context are often implicit. Therefore, we perform a extraction operation on the plain context to uncover the logical relations in it using a language model. The process can be defined as:

$$g = f_\theta(c, q, P_{in}) \tag{1}$$

where $f_\theta(\cdot)$ is a language model parameterized by $\theta$, $P_{in}$ is input task prompt. The specific task prompt $P_{in}$ used in our paper is displayed in the Appendix A. We perform Equation 1 for each $c$ in the sample space to obtain the extracted context space $\mathcal{G}$. We use the MindMap [10] structure to display the reorganized content, because it serves as a powerful structure for representing knowledge, concepts, and perspectives, contains not only logical relationships but also multi-hop connections.

As shown in Figure 2 (bottom left), the extracted context $g \in \mathcal{G}$ contains not only parallel logical relationships, e.g., Director&Producer, but also causal relationships, e.g., Julius Caesar→Director. Furthermore, it also describes the three-hop connections, such as Julius Caesar→Director→Name, Julius Caesar→ Director→Occupation, and Julius Caesar→Director→Education. This information with logical relationships enables LLMs to deeply understand the contextual content by clearly perceiving these logical relationships, facilitating the quality and reliability of reasoning. Additionally, this multi-hop connection corresponds to the complex multi-hop problem and therefore helps to solve this multi-hop question $q$.

### 3.2.2 Pruning

As described in Section 3.2, the extracted context $g \in \mathcal{G}$ contains various logical relationships and rich attributes. However, not all logical relationships and attributes help answer the question $q$. On the contrary, some may even interfere with the response to the question. For example, consider the question in Figure 2, the content "Julius Caesar → Production Company" is a distracting element, and "Julius Caesar → Adaptation" is irrelevant to the question.

To further reduce the interference of distracting or irrelevant logical relations and attributes on the retrieval of answers for question $q$, we use a pruning model trained through reinforcement learning (RL). The pruning model is based on the pre-trained BERT [11] due to its high generalizability, as shown in Figure 3. Its input consists of concatenated logical relationships, their corresponding attribute values from $g$, and the question $q$. For example, to prune the relation Julius Caesar → Adaptation, the input is: [CLS] Julius Caesar Adaptation Play by Shakespeare [SEP] Question [SEP]. We give a detailed demonstration of input format in Appendix B.

**RL Formulation** We formulate the pruning policy model optimization as an RL problem and employ proximal policy optimization (PPO) [23]. The action is keeping or deleting the logical relations. The policy decides the action probability given the question and $g$. We fine-tune the policy model $\pi$ by optimizing the reward $r$:

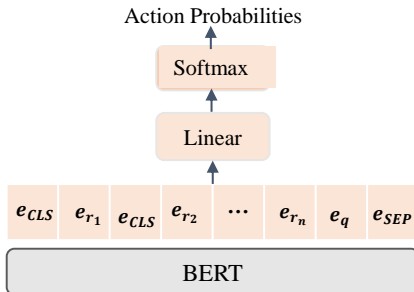

Figure 3: Illustration of Pruning model. The representation of [CLS] is used to obtain action probabilities.

$$\mathbb{E}_\pi[r] = \mathbb{E}_{g \sim \mathcal{G}, q \sim \mathcal{Q}, z \sim \pi(\cdot | x, q)}[r(z, q)] \tag{2}$$

**Reward Function** Our goal is to maximize LLM's generation toward the desired target by an alignment measure $\mathcal{R}$, and we use this as a reward. In this paper, the alignment metric we have

chosen is the F1 score. To keep the policy network $\pi$ from moving too far from the old result, we add a clipped term in the reward. Therefore, the final reward becomes:

$$r(\boldsymbol{z}, \boldsymbol{q}) = \min(\mathcal{R}(\boldsymbol{z}, \boldsymbol{q}), \mathrm{clip}(\pi(\boldsymbol{z} \mid \boldsymbol{x}, \boldsymbol{q}), 1 - \epsilon, 1 + \epsilon)) \tag{3}$$

where $\epsilon$ is a hyperparameter indicating the range of CLIP to be performed. After pruning, the extracted contextual content $g$ becomes the context $g^{'}$, which is closely related to the reasoning objective $q$. We perform a pruning operation for each $g \in \mathcal{G}$ to obtain the pruned context space $\mathcal{G}^{'}$.

### 3.3  Reasoning

After the re-organization process in Section 3.2, we get the re-organized context $g^{'} \in \mathcal{G}^{'}$. Then the re-organized context $g^{'}$ can be used as a context alone or combined with the original context $c$ to get the final answer of $q$. The reasoning process can be defined as:

$$o = f_\theta(g^{'}, [c], q, P_r) \tag{4}$$

where $P_r$ denotes prompt, and the content within $[]$ denotes that it is optional. The specific prompt used in reasoning is displayed in the Appendix C.

## 4  Experimental

### 4.1  Tasks and Datasets

To verify the effectiveness of our information re-organization method, we conduct experiments across a range of contextually aware multi-hop reasoning tasks and datasets, including claim verification [12], question answering [13], and reading comprehension [14]. The detailed dataset information, including data splits and statistics, is available in Appendix D.

**Claim Verification** The task involves assessing a given claim against a set of evidence documents to determine whether they support or refute the claim [12]. We consider HOVER [24] and FEVEROUS [25], which comprise complex claims that necessitate multi-hop reasoning for verification. Besides, we also take into account the SCIFACT [26] dataset, notable for its inclusion of scientific claims.

**Question Answering** For this task, we consider the following datasets: 2WikiMultiHopQA [27], StrategyQA [28], MuSiQue [29], and HotpotQA [30]. To answer the questions in these datasets requires not only multi-hop reasoning but also cross-document analysis.

**Reading Comprehension** Machine reading comprehension task requires a model to process documents and select an answer from the provided candidates to a question about the content [14]. We primarily consider WIKIHOP [31] in the task, which necessitates multi-hop reasoning to derive the final answer. Additionally, HotpotQA [30] is also frequently considered as part of the QA domain.

### 4.2  Baselines

In our paper, we compare our method InfoRE to two reasoning baselines: 1) **Standard**. 2) **CoT**. The Standard approach is a method that directly reasons with the original textual context. The CoT [6] method involves augmenting standard reasoning methods by adding a step-by-step thought process. In our paper, we conduct the CoT strategy by appending the sentence "Let's think step by step." at the end of the question.

The baseline methods and our InfoRE both adopt a zero-shot setting to counteract the potential randomness associated with demonstrations in a few-shot setting. We also design an answer-format instruction within the prompts for various tasks to standardize the structure of the final answer, thereby enhancing the precision of answer extraction. Moreover, all results reported in the paper use only the reorganized contextual information to reason. Comprehensive details about prompts and answer-format instruction are available in Appendix C. Following previous methods [17, 18, 25], we run the official evaluation scripts of each dataset to get the F1 to measure the results.

### 4.3  Implementation Details

In our paper, the LLMs employed in the extraction and reasoning process include Llama2-70B [2], GPT-3.5 (text-davinci-003) [32] and GPT-4 [3]. We use the official version of Llama2-70B. The

specific version of GPT-4 is GPT-4-0613. We configure all models with top_p parameter as 1.0 and temperature as 0.0. In the policy model, we use the BERT-base version on all tasks and datasets. In RL training, we calculate the F1 score between the generated answer and the reference answer as the reward, with a rescaling coefficient of 10. We train the model for 1000 episodes. We conduct training for epoch 5, a batch size of 4, and a learning rate of 2e-6. The parameter of $\epsilon$ is set to 0.2. All experiments are conducted on an NVIDIA RTX A6000.

## 5 Results and Analysis

### 5.1 Main Results

**Claim Verification** Table 1 presents a comprehensive performance comparison between our InfoRE and existing zero-shot techniques. For the HOVER dataset, we segment it into the 2-hop, 3-hop, and 4-hop levels following previous methods [17]. As depicted in Table 1, our InfoRE demonstrates significant improvements in the zero-shot claim verification task. The CoT [6] approach offers a lightly increase of 0.62% in the HOVER 4-hop using GPT-4, which indicates its marginal utility in more complex reasoning scenarios. Yet, our InfoRE achieves 3.02% improvement on the HOVER 4-hop using GPT-4 showing remarkable performance on contextual aware understanding and reasoning. This improvement is further increased to 73.62% in combination with CoT, suggesting that the methods complement each other effectively. In the case of Llama2-70B, the combined application of InfoRE and CoT yields a score of 53.20% on the 2-hop HOVER task, surpassing its CoT-only score of 50.02%. This pattern of improvement is consistent with GPT-3.5. GPT-4 shows superior performance across all methods and datasets, suggesting an inherent advanced reasoning ability. This performance is mirrored in the specialized benchmarks, where GPT-4 with InfoRE attains near-perfect accuracy on FEVEROUS (95.62%) and very high accuracy on SCIFACT (93.67%). These figures solidify the notion that the InfoRE indeed enhances the reasoning capabilities of LLMs.

Table 1: Zero-shot performance on claim verification task across three LLMs.

| LLMs | Methods | HOVER | | | FEVEROUS | SCIFACT |
|---|---|---|---|---|---|---|
| | | 2-hop | 3-hop | 4-hop | | |
| LLAMA2 (70B) | Standard | 49.41 | 48.35 | 47.82 | 63.39 | 60.70 |
| | InfoRE | **52.83** | **51.42** | **50.04** | **67.84** | **63.81** |
| | | ↑ 3.42 | ↑ 3.07 | ↑ 2.22 | ↑ 4.45 | ↑ 3.11 |
| | CoT | 50.02 | 48.76 | 48.01 | 64.53 | 61.24 |
| | InfoRE + CoT | **53.20** | **51.70** | **50.15** | **68.12** | **64.02** |
| | | ↑ 3.18 | ↑ 2.94 | ↑ 2.14 | ↑ 3.59 | ↑ 2.78 |
| GPT-3.5 | Standard | 64.74 | 63.04 | 61.54 | 87.67 | 77.42 |
| | InfoRE | **68.21** | **66.45** | **64.91** | **91.31** | **81.54** |
| | | ↑ 3.47 | ↑ 3.41 | ↑ 3.37 | ↑ 3.64 | ↑ 4.12 |
| | CoT | 66.70 | 64.52 | 62.69 | 88.67 | 78.49 |
| | InfoRE + CoT | **69.02** | **67.53** | **65.66** | **91.53** | **82.26** |
| | | ↑ 2.32 | ↑ 3.01 | ↑ 2.97 | ↑ 2.86 | ↑ 3.77 |
| GPT-4 | Standard | 72.40 | 71.02 | 70.06 | 92.33 | 91.40 |
| | InfoRE | **75.87** | **74.06** | **73.08** | **95.62** | **93.67** |
| | | ↑ 3.47 | ↑ 3.04 | ↑ 3.02 | ↑ 3.29 | ↑ 2.27 |
| | CoT | 73.82 | 72.07 | 70.68 | 92.67 | 92.47 |
| | InfoRE + CoT | **76.69** | **75.16** | **73.62** | **95.67** | **94.32** |
| | | ↑ 2.87 | ↑ 3.09 | ↑ 2.94 | ↑ 3.00 | ↑ 1.85 |

**Question Answer and Reading Comprehension** Different from claim verification task, this task involves using multiple documents as context, presenting a challenge in cross-document reasoning. Intuitively, when the reasoning process involves multiple documents, information re-organization can effectively merge the information from different documents and uncover logical relationships that are not apparent in plain text. Performance in Table 2 verifies the intuitive. We can see that after applying information re-organization, all the results have a significant improvement. GPT-4 outperforms

the other models, with the highest F1 score of 76.52%, 71.20%, and 83.22% when employing InfoRE on 2WikiMultiHopQA, StrategyQA, and HotpotQA, respectively. This is consistent with the performance improvements observed with GPT3.5 (text-davinci-003) and Llama2-70B when applying InfoRE. Different from conventional QA tasks, reading comprehension tasks require LLMs to not only deeply understand the context but also identify distractors among the candidates, increasing the reasoning challenge. Our InfoRE consistently shows improvements on this task. Specifically, in the WIKIHOP dataset, GPT-3.5 with InfoRE outperforms other methods with an F1 of 51.87%. This improvement further verifies the effectiveness of our method.

Table 2: Zero-shot results on Question Answering and Reading Comprehension tasks. 2WMHQA, SQA, and HQA are abbreviations for 2WikiMultiHopQA, StrategyQA, and HotpotQA, respectively.

| LLMs | Methods | 2WMHQA | MuSiQue | SQA | HQA | WIKIHOP |
|---|---|---|---|---|---|---|
| LLAMA2 (70B) | Standard | 52.56 | 49.55 | 51.23 | 66.07 | 40.32 |
| | InfoRE | **57.62** | **52.78** | **55.32** | **69.98** | **42.90** |
| | ↑ 5.06 | ↑ 3.23 | ↑ 4.09 | ↑ 3.91 | ↑ 2.58 |
| | CoT | 52.99 | 52.90 | 56.80 | 66.80 | 41.07 |
| | InfoRE + CoT | **57.72** | **56.10** | **59.93** | **70.60** | **43.37** |
| | ↑ 4.73 | ↑ 3.20 | ↑ 3.13 | ↑ 3.80 | ↑ 2.30 |
| GPT-3.5 | Standard | 58.25 | 55.01 | 59.39 | 73.30 | 48.92 |
| | InfoRE | **64.58** | **58.03** | **63.16** | **77.12** | **51.87** |
| | ↑ 6.33 | ↑ 3.02 | ↑ 3.77 | ↑ 3.82 | ↑ 2.95 |
| | CoT | 59.37 | 57.05 | 67.51 | 73.90 | 49.65 |
| | InfoRE + CoT | **65.13** | **60.52** | **70.45** | **77.74** | **52.70** |
| | ↑ 5.76 | ↑ 3.47 | ↑ 2.94 | ↑ 3.84 | ↑ 3.05 |
| GPT-4 | Standard | 72.69 | 62.65 | 68.32 | 79.33 | 55.46 |
| | InfoRE | **76.52** | **66.36** | **71.20** | **83.22** | **58.01** |
| | ↑ 3.83 | ↑ 3.71 | ↑ 2.88 | ↑ 3.89 | ↑ 2.55 |
| | CoT | 74.08 | 64.36 | 68.50 | 80.66 | 56.02 |
| | InfoRE + CoT | **78.60** | **69.11** | **71.54** | **84.26** | **58.91** |
| | ↑ 4.52 | ↑ 4.75 | ↑ 3.04 | ↑ 3.60 | ↑ 2.89 |

## 5.2 Analysis

**Ablation studies**    In our paper, the re-organization comprises two components: extraction and pruning. To investigate the impact of each component in detail, we conduct a series of ablation experiments using GPT-3.5 on the 2WikiMultiHopQA dataset. First, we directly remove extraction and pruning from our method, and the results are shown in the second and third rows of Table 3, respectively. It is worth mentioning that in the experiment where extraction is removed, we directly prune the sentences in the original context. Furthermore, we replace the reinforcement learning-based pruning method with a similarity-based pruning method to demonstrate its effectiveness. Specifically, the similarity-based pruning method uses Siamese-BERT, which takes the original question and each logical relationship as inputs separately, and then generates the corresponding representations. Then, we calculate the cosine similarity between these two representations. Finally, we removed the 30% of logical relationships with the lowest similarity, the result is shown in the last row of Table 3.

The results in Table 3 show that removing the extraction and pruning operations leads to performance drops of 2.94% and 1.53%, respectively. This demonstrates the effectiveness of both components in our methods. The larger performance drop after removing extraction highlights the importance of extracting logical relationships for effective reasoning. After replacing the pruning model, the performance dropped by 1.26%, but the results were still better than without pruning.

Table 3: F1 performance of ablation studies.

| Methods | 2WikiMultiHopQA |
|---|---|
| **Full model** | **64.58** |
| w/o extraction | 61.64 |
| w/o pruning | 63.05 |
| similarity-based pruning | 63.32 |

This not only demonstrates the necessity of pruning but also highlights the effectiveness of the reinforcement learning-based pruning method.

**Quality of Re-Organized Information**    To assess the quality of the re-organized information, we perform a quantitative evaluation of the re-organized information with GPT-4 (gpt-4-32k) on the 2WikiMultiHopQA dataset. Specifically, we select 100 samples from the dataset, GPT-4 (gpt-4-32k) is asked to rank re-organized information produced by GPT-3.5 (text-davinci-003) [1] and GPT-4, as well as original context following criteria: (1) Depth: The information present multiple relationships of a topic, offering insightful perspectives or in-depth understanding of the subject. (2) Clarity: Information is clear and precise, making it easy to understand without ambiguity. All of the information is ranked 1, 2, and 3 with 3, 2, and 1 scores, respectively. Finally, we get a weighted average score for each information to measure the overall quality.

The results in Table 4 demonstrate that the re-organized information outperforms the original textual context in terms of depth and clarity, which justifies the motivation of our paper. In addition, the re-organized information from GPT-4 outperforms GPT-3.5, which proves in another way that GPT-4 is indeed more capable than GPT-3.5. The evaluation results further validate the effectiveness of our method. Furthermore, the 22.22% improvement in depth is more obvious than the 15.14% improvement in clarity, which indicates that the re-organization of the information has been particularly effective in enhancing the logical relationships of the information. The improvement in clarity suggests that the information is now presented in a more direct and streamlined manner, making it easier for LLMs to grasp the essential points without wading through unnecessary details.

Table 4: Qualitative evaluation results on 2Wiki-MultiHopQA dataset. Avg R denotes the weighted average ranking score. The larger ranking score denotes better information quality.

| Methods | Depth | | | |
| --- | --- | --- | --- | --- |
| | 1st | 2nd | 3rd | Avg R. |
| Original | 0.22 | 0.36 | 0.42 | 1.80 |
| GPT-3.5 | 0.32 | 0.36 | 0.32 | 2.00 |
| **GPT-4** | 0.46 | 0.28 | 0.26 | **2.20** |

| Methods | Clarity | | | |
| --- | --- | --- | --- | --- |
| | 1st | 2nd | 3rd | Avg R. |
| Original | 0.25 | 0.35 | 0.40 | 1.85 |
| GPT-3.5 | 0.35 | 0.32 | 0.33 | 2.02 |
| **GPT-4** | 0.40 | 0.33 | 0.27 | **2.13** |

**Effect of Re-organized Information Quality**

To explore the effects of the quality of re-organized context on model reasoning capabilities, we utilize both GPT-3.5 (text-davinci-003) and GPT-4 to re-organize context information. Reasoning processes are then independently executed on each model. Moreover, employing this cross-validation technique also allows us to effectively evaluate the robustness of our method.

The cross-validation results, shown in Table 5, indicate that the use of either GPT-3.5 or GPT-4 for information re-organization leads to a marked improvement in the performance of LLMs on reasoning tasks. Additionally, it is observed that using GPT-4 for information re-organization while reasoning with the GPT-3.5 results in a further increase in reason results by 1.19% and 2.03% on FEVEROUS and 2WikiMultiHopQA datasets, respectively. This implies that enhancing the quality of re-organized information can improve the performance of LLMs, pointing to a promising direction for future research in refining in-

Table 5: F1 performance of cross-validation, where InfoRE* denotes reason with GPT-3.5 but information re-organization with GPT-4, InfoRE† denotes reason with GPT-4 but information re-organization with GPT-3.5 (text-davinci-003).

| Methods | FEVEROUS | 2WikiMultiHopQA |
| --- | --- | --- |
| GPT-3.5 (†) | | |
| Standard | 87.67 | 58.25 |
| InfoRE | 91.31 | 64.58 |
| **InfoRE** * | **92.50** | **66.61** |
| GPT-4 (*) | | |
| Standard | 92.33 | 72.69 |
| **InfoRE** | **95.62** | **76.52** |
| InfoRE† | 94.67 | 75.07 |

formation synthesis and re-organization. Conversely, when GPT-3.5 is employed for information re-organization in conjunction with reasoning using the GPT-4 model, there is a decrease in reason ability by 0.95% and 1.45% respectively. In addition, the magnitude of the decrease is lower than the increase, which implies that our re-organization strategy may have a greater impact on models with

weaker reasoning abilities. This finding aligns with our understanding that models with inherently weaker reasoning abilities tend to rely more heavily on external strategy. For models with stronger inherent capabilities, our method still further improve its reasoning ability.

**Error Analysis**   To better comprehend where the errors in our InfoRE methodology come from and where they are fixed, we annotate 100 wrong predictions made by both InfoRE and Standard methods with GPT-3.5 on 2WikiMultiHopQA dataset. We categorize the errors into 4 classes:

- **Contextual Misunderstanding (CM)**: This happens when the model fails to interpret or connect multiple pieces of information from different parts of the documents. Multi-hop reasoning requires synthesizing information from various segments, and recognizing logical relations, and any misunderstanding can lead to incorrect conclusion.

- **Factual Error (FE)**: The model may provide an answer that is factually incorrect or not supported by the given documents. This is often due to the model's reliance on its training data, which may not always align with the specific facts in the context.

- **Mathematical Error (ME)**: The error occurs when math calculations are involved in deriving the final answer.

- **Unanswerable Question (UQ)**: It's a specific type of error or limitation in dataset design, where the context does not contain enough information to provide a valid answer to the posed question.

When engaging in reasoning with LLMs, all four error categories are present. As shown in Figure 4, there are 6% unanswerable question errors in the dataset, and more than 90% errors occur in the reasoning in the baseline method. Among the four types of error, contextual misunderstanding is the primary source of errors in the baseline, highlighting the importance of an in-depth understanding of context in solving reasoning tasks with LLMs. This finding is consistent with our motivation presented in the introduction section. Moreover, comparing the results of errors between our method and the baseline method, our method mainly corrects 14% of errors coming from the baseline method, most of the corrected errors are contextual misunderstanding errors. This further indicates that our InfoRE method assists LLMs in understanding context, signifying the necessity and effectiveness of conducting information re-organization before directly addressing the original question.

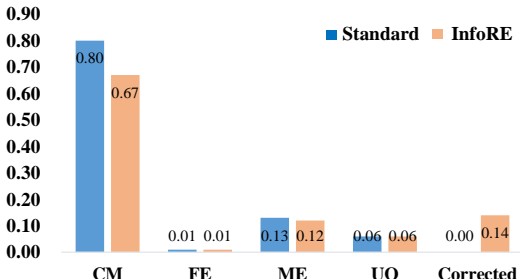

Figure 4: Error Analysis of InfoRE on 2Wiki-MultiHopQA against Standard baseline method. The first four rectangles are error categories, while "Corrected" on the far right denotes the percentage of errors originally made by the baseline method that our method InfoRE has successfully corrected.

## 6   Conclusion

In this paper, we propose an information re-organization method to improve the reasoning ability of LLMs. Compared with previous approaches primarily focus on improving the quality of intermediate steps, our method emphasizes uncovering the logical relationships, multi-hop connections, and pruning the irrelevant information through information re-organization. This approach enables LLMs to explicitly perceive the logical relationships and multi-hop connections of concepts within the context, promoting a deeper integration and understanding of the context, which results in more robust reasoning outcomes. To verify the effectiveness of our method, we conduct experiments using various LLMs across a range of contextually aware multi-hop reasoning tasks. The experiment results demonstrate the potential of our method to improve the reasoning ability of LLMs. Additionally, our method has a positive impact on various tasks involving context understanding, such as academic research, legal analysis, and medical diagnostics. However, it is also important to be aware of potential negative impacts, such as the propagation of misinformation.

## Acknowledgments and Disclosure of Funding

This work is supported by the National Natural Science Foundation of China (No. 62376245), the "Pioneer" and "Leading Goose" R&D Programs of Zhejiang (No. 2024C03255), the Fundamental Research Funds for the Central Universities (226-2024-00170), the project of the Donghai Laboratory (Grant no. DH-2022ZY0013), National Key Research and Development Project of China (No. 2018AAA0101900), and MOE Engineering Research Center of Digital Library.

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

# A Prompt for Extraction

The specific prompt of extraction is shown in Table 6. The purpose of the prompt is to obtain logical relationships and multi-hop connections from the context. In the prompt, the content within the brackets should be replaced with specific example content.

Table 6: Specific prompt for obtaining logical relationships. During execution, "[EVIDENCE]" needs to be replaced with the specific document, and "[CLAIM]" is replaced with the specific question.

| Prompt Content for Logical Relationship Extraction |
|---|
| Given a claim and corresponding evidence, please summarize the evidence as a mind map according to the claim. The output must be in a strict JSON format: {"mind_map": "mind_map"}. CLAIM: [CLAIM] EVIDENCE: [EVIDENCE] |

To clearly demonstrate the content of the LLM after performing the extraction, we give an example in the MuSiQue dataset.

Document: Julius Caesar is a 1953 epic Metro-Goldwyn-Mayer film adaptation of the play by Shakespeare, directed by Joseph L. Mankiewicz, who also wrote the uncredited screenplay, and produced by John Houseman. The original music score is by Miklos Rozsa. The film stars Marlon Brando as Mark Antony, James Mason as Brutus, John Gielgud as Cassius, Louis Calhern as Julius Caesar, Edmond O'Brien as Casca, Greer Garson as Calpurnia, and Deborah Kerr as Portia. Houseman was born on September 22, 1902, in Bucharest, Romania, the son of May (Davies) and Georges Haussmann, who ran a grain business. His mother was British, from a Christian family of Welsh and Irish descent. His father was an Alsatian-born Jew. He was educated in England at Clifton College, became a British subject, and worked in the grain trade in London before emigrating to the United States in 1925, where he took the stage name of John Houseman. He became a United States citizen in 1943.

JSON format after extraction:

```
{
  "Julius Caesar": {
    "Production Company": "Metro-Goldwyn-Mayer",
    "Adaptation": "Play by Shakespeare",
    "Director": "Joseph L. Mankiewicz",
    "Screenplay": {
        "Writer": "Joseph L. Mankiewicz",
        "Credit": "Uncredited"
        },
    "Producer": {
        "John Houseman": {
            "Birth": {
                "Date": "September 22, 1902",
                "Place": "Bucharest, Romania"
            },
            "Parents": {
                "Mother": {
                    "Name": "May Davies",
                    "Nationality": "British",
                    "Religion": "Christian",
                    "Descent": "Welsh and Irish"
                },
                "Father": {
                    "Name": "Georges Haussmann",
                    "Occupation": "Grain Business",
                    "Religion": "Jew",
```

```
                "Birthplace": "Alsace"
            }
        },
        "Education": "Clifton College, England",
        "Citizenship": {
            "Original": "British",
            "Naturalized": "United States, 1943"
        },
        "Occupation": {
            "Original": "Grain Trade, London",
            "Stage Name": "John Houseman"
        },
        "Emigration": "United States, 1925"
    }
},
"Music Score": {
    "Composer": "Miklos Rozsa"
},
"Cast": {
    "Marlon Brando": "Mark Antony",
    "James Mason": "Brutus",
    "John Gielgud": "Cassius",
    "Louis Calhern": "Julius Caesar",
    "Edmond O'Brien": "Casca",
    "Greer Garson": "Calpurnia",
    "Deborah Kerr": "Portia"
}
    }
}
```

## B  Input Format For Pruning

The extracted context $g$ includes logical relationships and corresponding attribute values. First, we iterate through all logical relationships and attribute values, and following the previous method [33], we concatenate them using [SEP] [CLS], combining them with the question to input into the pruning model. Then, we use the representation of the [CLS] token to represent the logical relationships for subsequent operations.

We use an example in Figure 2 to demonstrate the input format of the extracted context $g$ in the pruning model. The question is: Where did the producer of Julius Caesar study or work? The the extracted context $g$ is:

```
Julius Caesar:
  Production Company: Metro-Goldwyn-Mayer
  Director: Joseph L. Mankiewicz
  Producer:
    Name: John Houseman
    Education: Clifton College, England
    Occupation: Grain Trade, London
  Adaptation: Play by Shakespeare
```

We only traverse logical relationships of the first-level progressive type. Therefore, the logical relationships contained in extracted context $g$ of example include 1) Production Company: Metro-Goldwyn-Mayer, 2) Director: Joseph L. Mankiewicz, 3) Producer: Name: John Houseman, 4) Producer: Education: Clifton College,England, 5) Producer: Occupation: Grain Trade, London, 6) Adaptation: Play by Shakespeare. Then, we concatenate them using [SEP][CLS].

The input format after concatenation is: [CLS] Production Company: Metro-Goldwyn-Mayer [SEP] [CLS] Director: Joseph L. Mankiewicz [SEP] [CLS] [CLS] Producer: Name: John Houseman [SEP]

[SEP] [CLS] Producer: Education: Clifton College [SEP] [CLS] Producer: Occupation: Grain Trade, London [SEP] [CLS] Adaptation: Play by Shakespeare [SEP] [CLS] Where did the producer of Julius Caesar study or work? [SEP].

## C   Prompt for Multi-hop Reason

During the reasoning stage using large language models, to accommodate more context-aware reasoning tasks while ensuring comparability of results, we designed a universal prompt template. The prompt template consists of three components: original context, e.g., documents or paragraphs, reorganized information, and a question. The prompt template is as follows:

```
Documents:
    [Re-Organized TEXT]
    [TEXT][Optional]
Question:
    [QUESTION]
please answer the question based on the documents.
Answer:
```

The specific prompts for Standard, chain-of-thought (CoT), our InfoRE, and InfoRE + CoT to reason are shown in Table 7. In the prompt, the content within the brackets [] should be replaced with specific example content.

Table 7: Specific prompts and answer format instructions of Standard, CoT, InforRE, and InfoRE + CoT in our paper. During execution, "[EVIDENCE]" needs to be replaced with the specific document, "[CLAIM]" is replaced with the specific question, and "[MindMap]" is replaced with the specific MindMap.

| Methods | Prompt Content |
| --- | --- |
| Standard | Documents: [EVIDENCE]
Question: [CLAIM]?
Please answer the question based on Documents.
Your final answer should be enclosed in XML tag <answer></answer>, like this:
<answer>{*final_answer*}</answer>, at the end of your response.
Answer: |
| CoT | Documents: [EVIDENCE]
Question: [CLAIM]?
Please answer the question based on Documents.
Your final answer should be enclosed in XML tag <answer></answer>, like this:
<answer>{*final_answer*}</answer>, at the end of your response.
Let's think step by step.
Answer: |
| InfoRE | Documents: [MindMap]
Question: [CLAIM]?
Please answer the question based on Documents.
Your final answer should be enclosed in XML tag <answer></answer>, like this:
<answer>{*final_answer*}</answer>, at the end of your response.
Answer: |
| InfoRE + CoT | Documents: [MindMap]
Question: [CLAIM]?
Please answer the question based on Documents.
Your final answer should be enclosed in XML tag <answer></answer>, like this:
<answer>{*final_answer*}</answer>, at the end of your response.
Let's think step by step.
Answer: |

# D    Detailed Dataset Information

In our experiments, due to the resource limitations of large language models, we sample a portion from each dataset, following previous methods [6, 16].

Table 8:   Details statics information of evaluation datasets we used in the paper. Pairs denote the number of examples. 2WMHopQA is the short name of 2WikiMultiHopQA.

| Tasks | Dataset | Pairs | |
|---|---|---|---|
| | | train | test |
| Claim Verification | FEVEROUS | 2000 | 2959 |
| | HOVER | 2000 | 4000 |
| | SCIFACT | 200 | 212 |
| Question Answering | 2WikiMultiHopQA | 2000 | 500 |
| | StrategyQA | 1000 | 229 |
| | MusiQue | 2000 | 2417 |
| Reading Comprehension | HotpotQA | 2000 | 500 |
| | WIKIHOP | 2000 | 500 |

**Claim Verification**    The task involves assessing a given claim against a set of evidence documents to determine whether they support or refute the claim [12]. We consider HOVER [24] and FEVEROUS [25], which comprise complex claims that necessitate multi-hop reasoning for verification. Besides, we also take into account the SCIFACT [26] dataset, notable for its inclusion of scientific claims.

**Question Answering**    For this task, we consider the following datasets: 2WikiMultiHopQA [27], StrategyQA [28], MuSiQue [29], and HotpotQA [30]. To answer the questions in these datasets requires not only multi-hop reasoning but also cross-document analysis.

**Reading Comprehension**    Machine reading comprehension task requires a model to process documents and select an answer from the provided candidates to a question about the content [14]. We primarily consider WIKIHOP [31] in the task, which necessitates multi-hop reasoning to derive the final answer. Additionally, HotpotQA dataset [30] is also frequently considered as part of the question answering domain.

# E    Limitations

We propose an information re-organization approach to improve the reasoning of large language models, which performs well on some context-aware reasoning tasks but still has some limitations. Firstly, the structures of information re-organization are limited. Generally, there are multiple structures for information reorganization, such as tables, timelines, etc. Next, we will extend the re-organization structure to include more types. Secondly, the re-organization process relies on large language models. If we can implement this re-organization using smaller language models, our method will become more generalizable. This is another direction we need to focus on in the future.

