# OpenReview forum: "Information Re-Organization Improves Reasoning in Large Language Models"
_NeurIPS.cc/2024/Conference — NeurIPS 2024 poster_

### Official Review · Reviewer_x5hg · 2024-07-06

**Soundness:** 3
**Presentation:** 3
**Contribution:** 3
**Rating:** 7
**Confidence:** 3

**Summary:**

The paper proposes a simple yet effective method to work with most of the current reasoning strategies. It automatically organizes the content into structural form, excluding noises and unused information during this process. The experiments are done using three models across ten datasets. The method consistently improves a vanilla setting and also a CoT setting. Ablation study shows the effectiveness of both the extraction part and the pruning part.

**Strengths:**

1.	The method is compatible with other reasoning strategies such as CoT and can further improve them. The method is easy to follow implement.
2.	The paper includes extensive experiments. The proposed method is verified on all the models (3 LLMs) across all selected tasks (10 datasets) and shows consistent improvement in all scenarios.
3.	The ablation study shows that each of the two components in the proposed method can improve the overall performance independently.

**Weaknesses:**

1.	Do you consider trying gpt-3.5-0613/1106/0125 instead of text-davinci-003? The text-davinci-003 model is no longer usable in OpenAI APIs.
2.	Some rationales need further explanation in the design of the proposed method. Do you consider using LLM for the information pruning part instead of a pre-trained BERT? Will there be any improvement if using LLM? Do you consider using an end-to-end model to do extraction and pruning at the same time?
3.	It is great to see those analysis in section 5.2. It will greatly improve the insight of your paper if you can include:
a.	Will the extraction process introduce some hallucination?
b.	The paper focuses on zero-shot setting currently. How can few-shot setting help? It may help from two perspectives: (1) Better information extraction; (2) Better downstream task performance.
c.	The paper uses a very simple reasoning technique, the CoT prompting. Will the proposed method help in more advanced on such as ToT?
4.	I am particularly interested in the aspect of minimizing noise. However, currently the experiments are done on dataset with normal data without many noises. Could you please try on something like web-crawled data? Another choice can be manually injecting some noises in the datasets you use, i.e., adding some meaningless tags (<a></a>), or some irrelevant sentences. I am interested to see how the proposed method can automatically exclude these noises.
5.	Also, since the method can formulate the content in a structural data. Could you please try something like changing the order of sentences in the content? Will it affect how the method extract data?

Minor suggestions to presentation:

1.	A space is missing between “2” and “(” at line 125.
2.	The “Large language models (LLMs)” appears at line 22, line 75, and line 353. Please do the abbreviation once at the beginning and use the abbreviation directly afterwards. Same suggestion applies to others like “Chain of Thought (CoT)” at line 190.
3.	There seems to be an extra period in the “Prompt Content for Logical Relationship Extraction” in appendix A.

**Questions:**

1.	Why do you name the method RE-organization? I don’t see any other organization process before you’re the process of your method in the pipeline. Will it be more proper to name it “Information Organization?”
2.	What version of LLaMA-2 are you using? The temperature of the hugging face version needs to be a positive value so it cannot be zero.
3.	Line 317-319: Is it also because GPT-4 has a better ability to do the re-organization job than GPT-3?
4.	Are the 100 wrong predictions you annotated randomly sampled?
5.	Could you please give some mind map examples that LLM output in JSON format? Probably you can append them in the Appendix for a clear illustration.

**Limitations:**

Yes.

---

> ### Author Rebuttal · Authors · 2024-08-06
>
> Many thanks to the reviewer for giving these valuable feedback and comments.
>
> **W1**: We supplement the results across all datasets using GPT-35-Turbo-0613, including the main experimental results and ablation results. The results are shown in Tables 1, 2, and 3 in the PDF file under the Author Rebuttal section.
>
> **W2**: We chose pre-trained BERT for information pruning due to its high generalizability. In our experiments using LLAMA-2-70B, GPT-3.5, and GPT-4 for the pruning on 2WikiMultiHopQA, only GPT-3.5 and GPT-4 surpass the current method. However, we aim to develop a foundational and general method, not one limited to specific conditions. Additionally, we are concerned about self-preference bias in LLMs. As Arjun Panickssery et al. noted in "LLM Evaluators Recognize and Favor Their Own Generations," LLMs tend to favor their own outputs. Using LLMs for both extraction and pruning could intensify this bias, making it hard to determine if improvements are due to methodological enhancements or bias.
>
> An end-to-end approach combining extraction and pruning was tested in our method. As shown in Table 6 of Appendix A, our prompt for context extraction included the specific question to filter unrelated content. However, the ablation study in Table 3 shows that additional pruning improves performance, indicating that simultaneous extraction and pruning is less effective than our current separate method.
>
> We will supplement the rationale analysis in the revised version of the paper to enhance the rationale of the approach.
>
>
> **W3**: a. LLM-generated content often contains hallucinations. In our experiments, we used GPT-4 to verify the factual consistency of the extracted information. If inconsistencies were found, we re-extracted the information until it was consistent. According to our statistics, 98% of the extraction results are consistent with the original text after the first extraction.
>
> b. The context of the dataset used in our experiments is relatively long. To prevent the text length from exceeding the maximum limit of LLMs, we adopt a 1-shot setting during the information extraction on the 2WikiMultiHopQA dataset, followed by direct reasoning with GPT-4. This approach resulted in an F1 score of 74.67%, which is lower than the existing zero-shot information extraction results of 76.52%. Considering the diversity of the samples in the dataset, using the 1-shot setting limited the effectiveness of the information extraction.
>
> c. To explore whether our method contributes to the ToT method, we conduct experiments with GPT-4 on all datasets. Specifically, we use GPT-4 to decompose a multi-hop question into a tree in BFS order, where the root node represents the original question and the other nodes represent sub-questions. The results are shown in Tables 1 and 2 in the PDF file under the Author Rebuttal section. The results show that our method, combined with ToT, can further enhance reasoning capabilities.
>
>
> **W4**: Following the reviewer's suggestion, we test our method on a noisier dataset constructed using retrieval tools. Due to time constraints, this is validated only on the 2WikiMultiHopQA dataset. Specifically, we use BM25, implemented with the Pyserini toolkit, to retrieve the top-3 paragraphs from the October 2017 Wikipedia corpus as context documents. We then apply our method and conduct the ablation study as described in section 5.2 using GPT-4. The results indicate that when the context contains more noise, the pruning operation brings a significant 2.55% improvement. The specific ablation results are as follows:
> | Methods | 2WikiMultiHopQA |
> | --- | --- |
> | Standard |  43.70 |
> | Full model (InfoRE) |  48.82 |
> | w/o extraction |  46.14($&darr; 2.68$ ) |
> | w/o pruning |  46.27($&darr; 2.55$ ) |
>
> **W5**: If there are no dependencies between sentences, changing their order will not affect the extraction results. Conversely, if there are dependencies, such as reference relationships, changing the sentence order will affect the extraction results.
>
> **presentation suggestions**:
>
> Thanks to the reviewer for the presentation suggestions, we will address them in the revised version.
>
> **Q1**: We chose the term "Re-Organization" to emphasize the characteristics of reintegration and optimization of the information process of our method. Although no prior organization process is explicitly mentioned before the pipeline, we consider that the raw data processing and analysis are usually organized and processed in some initial form. Therefore, our method further "re-organizes" this to enhance the relational richness and usability of information.
> However, we greatly appreciate your suggestion to use the name "Information Organization" to more intuitively reflect the function of the method. We will consider this change to more accurately describe our method.
>
> **Q2**: We use the official version of the model. Specifically, we register on the Meta website (https://llama.meta.com/llama-downloads/) to download the model. Following the usage example in example_chat_completion.py on the official Llama GitHub (https://github.com/meta-llama/llama), we set the temperature to 0.
>
> **Q3**: More specifically, it is because GPT-4 inherently has stronger capabilities than GPT-3, whether in terms of organizing information or reasoning.
>
> **Q4**: Yes, these 100 samples are randomly selected from our results of prediction errors. This random sampling ensures an unbiased analysis of the error types our model is making, providing us with a comprehensive view of the potential areas for improvement.
>
> **Q5**: Due to the rebuttal length limit of 6000 tokens, we do not have enough space to display the JSON format mind map requested by the reviewer. However, the JSON format mind map we used is very similar to the example shown in Figure 2 of the paper, except that the example in Figure 2 omits the curly braces "{}".
> More samples will be included in the appendix of the revised version for a clear illustration.

---

> > ### Comment · Reviewer_x5hg · 2024-08-13
> >
> > Thanks very much. Most of my concerns have been adressed. I raise my score by one point.

---

> ### Author Response · Authors · 2024-08-13
> **Appreciation for Positive Feedback**
>
> Dear Reviewer x5hg,
>
> Thank you for your positive reassessment. We greatly appreciate your recognition and insightful suggestions.  We will incorporate suggestions and additional analysis results into the revised paper to enhance its completeness and clarity.
>
> Once again, thank you for your time and effort in reviewing our paper. If you have any further questions, please feel free to reach out at any time.
>
> Best regards,
>
> The authors of InfoRE

---

### Official Review · Reviewer_cK3D · 2024-07-10

**Soundness:** 3
**Presentation:** 3
**Contribution:** 2
**Rating:** 5
**Confidence:** 4

**Summary:**

This paper proposes a method called information re-organization (InfoRE) to enhance the performance of large language models on some reasoning tasks. Unlike existing approaches that primarily focus on refining the reasoning process, InfoRE emphasizes re-organizing contextual information to identify logical relationships before reasoning. This method involves extracting logical relationships from the context and pruning redundant content to minimize noise. The re-organized information is then used in the reasoning process. The authors demonstrate the effectiveness of InfoRE through experiments on several contextually aware multi-hop reasoning tasks, achieving an average improvement of 4% in a zero-shot setting.

**Strengths:**

- The idea of information re-organization is intuitive.
- The demonstrated improvement in reasoning performance by an average of 4% across tasks, as well as the ablation study, verifies the functionality of the proposed method.
- This paper is easy to follow.

**Weaknesses:**

- The idea of this paper can be connected to existing prompt engineering works, e.g., performing retrieval from context and query/context rewriting. I feel this paper lacks technical depth (though this may be the fashion of a prompt engineering work, i.e., insightful but lacking technical depth). And the insight is not significant enough to compensate for the limited technical depth.
- Though a 4% performance improvement may be considered significant, the baseline in this paper is quite weak (e.g., standard prompting or vanilla CoT). More baselines may need to be included and compared.
- From the limited ablation study on 2WikiMultiHopQA, we can observe that the improvement from pruning is not significant. This makes me doubt whether such a complex/expensive design is really necessary.

**Questions:**

See weaknesses.

**Limitations:**

I don't see potential negative societal impact of this work.

---

> ### Author Rebuttal · Authors · 2024-08-06
>
> Thanks for reviewing our work and providing these valuable comments.
>
> **W1**: Our paper introduces a strategy specifically designed to enhance multi-hop reasoning capabilities by effectively organizing contextual information before reasoning processes begin. This approach addresses a noticeable gap in the literature, which traditionally emphasizes intermediate reasoning steps without sufficient focus on the initial organization of relevant contexts. Our method is simple yet effective and reflects a deliberate design choice aimed at maximizing efficiency and applicability across various tasks and datasets. We have rigorously tested our approach on multiple datasets, demonstrating its effectiveness in improving reasoning outcomes, with substantial improvements over baseline methods.
>
> Contrary to existing works that focus primarily on query rewriting (Query Rewriting for Retrieval-Augmented Large Language Models, EMNLP 2023) to better retrieve context that is more relevant to the query, and context rewriting (Learning to Compress Prompts with Gist Tokens, NeurIPS 2023) aims to compress the length of the context, our approach innovates by post-processing context to extract and prune information precisely. This fine-grained manipulation of context is both technically challenging and critical for the nuanced understanding required in multi-hop reasoning tasks.
>
> Furthermore, we focus on developing a foundational precedent for future developments in large language model reasoning processes. The practical implications of our research offer a scalable solution that can be integrated with existing and future LLM frameworks to facilitate more robust reasoning capabilities.
>
> We hope that this clarification addresses your concerns and further illustrates the technical rigor and innovative insights our work contributes to the field of prompt engineering.
>
>
> **W2**: Firstly, while the Chain of Thought (CoT) method is indeed simpler than some other methods, it has consistently demonstrated its effectiveness across a variety of reasoning tasks. Its widespread use and acknowledged efficacy in the research community make it an appropriate baseline for evaluating novel methodologies like ours. We aim to build upon well-established foundations to ensure that our contributions are both measurable and meaningful.
> Secondly, following the reviewer's suggestion, we add a stronger baseline method Tree-of-Thoughts (ToT). According to the original Tree-of-Thoughts paper, we adopt BFS search to solve the complex question. Specifically, we use GPT-4 to decompose a multi-hop question into a tree in BFS order, where the root node is the original question, and the other nodes are sub-questions. The specific results are as follows:
>
> | Model | Methods | HOVER-2 | HOVER-3 | HOVER-4 | FEVEROUS | SCIFACT |
> | --- | --- | --- | --- | --- | --- | --- |
> | GPT-4 | ToT | 74.36 | 72.43 | 70.64 | 92.78 | 92.33 |
> | GPT-4 | ToT+InfoRE | 76.96 | 74.18 | 72.30 | 95.82 | 94.42 |
>
> | Model | Methods | 2WMHQA | MuSiQue | SQA | HQA | WIKIHOP |
> | --- | --- | --- | --- | --- | --- | --- |
> | GPT-4 | ToT | 75.32 | 65.12 | 68.96 | 81.73 | 57.46 |
> | GPT-4 | ToT+InfoRE | 79.45 | 69.57 | 72.28 | 85.26 | 60.24 |
>
> The inclusion of the Tree-of-Thought baseline has provided a more challenging comparison, and our method still demonstrates nearly a 4% improvement in performance on the same datasets, reinforcing the effectiveness and robustness of our approach.
>
> **W3**:  Our current experiments are done on normal data without much noise. This is why the impact of the pruning operation is not very pronounced, although it still results in a 1.53% improvement on the 2WikiMultiHopQA dataset. To further demonstrate the significance of the pruning effect, we use a retrieval tool to construct contexts with more noise for the 2WikiMultiHopQA dataset, replacing the currently used golden contexts. Specifically, we use BM25, implemented with the Pyserini toolkit, to retrieve the top-3 paragraphs from the October 2017 Wikipedia corpus as context documents. We then apply our method on this dataset and conduct the ablation study as described in section 5.2 using GPT-4. From the ablation study results, we observe a notable increase in performance. Specifically, in these noisier conditions, the pruning operation resulted in a significant improvement of 2.55%, which is comparable with the ability of extraction.
> The ablation study results are as follows:
>
> | Methods | 2WikiMultiHopQA |
> | --- | --- |
> | Standard |  43.70 |
> | Full model (InfoRE) |  48.82 |
> | w/o extraction |  46.14($&darr; 2.68$ ) |
> | w/o pruning |  46.27($&darr; 2.55$) |

---

> > ### Author Response · Authors · 2024-08-13
> > **A Gentle Reminder**
> >
> > Dear Reviewer cK3D,
> >
> > Thank you for your efforts in reviewing our paper. We greatly value the feedback from each reviewer and have provided detailed responses to the concerns raised. Up to this point, we have received feedback from reviewers wQkW and x5hg, and are pleased to have their positive recognition. However, we have not yet received any feedback from you.
> >
> > As the discussion period is nearing its end, we would greatly appreciate it if you could let us know whether we have adequately addressed your concerns. Your insights are important to us, and we believe they will help us further improve the quality of our paper.
> >
> > Thank you once again for your time and consideration.
> >
> > Best regards,
> >
> > The authors of InfoRE

---

> > > ### Comment · Reviewer_cK3D · 2024-08-13
> > >
> > > Thank you for your response. It solves some of my concerns. I have updated my score.

---

> > > > ### Author Response · Authors · 2024-08-13
> > > > **Appreciation for Reassessment**
> > > >
> > > > Dear Reviewer cK3D,
> > > >
> > > > Thank you for taking the time to reassess our work. We sincerely appreciate your patience and professionalism throughout the entire review process, which have been instrumental in helping us further improve our research.
> > > >
> > > > Once again, thank you for your time and effort in reviewing our paper. If you have any further questions, please feel free to reach out at any time.
> > > >
> > > > Best regards,
> > > >
> > > > The authors of InfoRE

---

### Official Review · Reviewer_wQkW · 2024-07-13

**Soundness:** 3
**Presentation:** 3
**Contribution:** 3
**Rating:** 7
**Confidence:** 4

**Summary:**

The paper introduces an "Information Re-organization" (InfoRE) method aimed at improving the reasoning abilities of large language models (LLMs). It highlights the deficiencies of existing approaches that focus on intermediate reasoning steps without adequately addressing the preliminary organization of contextual information. The proposed InfoRE method involves extracting logical relationships and pruning redundant content from contextual data, guiding the reasoning process. This restructuring allows LLMs to recognize and use these logical relationships effectively, potentially enhancing reasoning quality. The method's efficacy is demonstrated using various LLMs like GPT-3.5 and GPT-4 across multiple reasoning tasks, showing an average improvement of 4% in performance.

**Strengths:**

1. The paper introduces a novel method of restructuring information before reasoning, which could be foundational for future developments in LLM reasoning processes.
2. Results indicate that InfoRE significantly improves reasoning tasks across multiple datasets and model architectures.
3. The paper provides a comprehensive breakdown of the method's components and their contributions to the overall performance.

**Weaknesses:**

1. Information extraction and pruning might require intricate tuning and be computationally expensive, making it less practical for real-time applications.
2. The paper is well structured. However, authors should provide more details about their experiments.

**Questions:**

1. What are the differences between your extraction method and extracting to knowledge graph triples? Leveraging Structured Information ... (Li et al., EMNLP 2023) proposed a method to extract knowledge graph triples from documents. Moreover, you should add this paper to your related work since it also extracts knowledge from the document.
2. Since generations from LLMs always change, can you provide a standard variance or significant score of your results to show your improvement is substantial?
3. Do you limit the type of entities and relations during extraction? Do you apply open relations or closed relations during extraction? I only find the prompt in Appendix A. Is this all about your extraction part?
4. How do you check the completeness and quality of your extraction results? Can you provide more details about this?
5. LLM answers are always long, even with the format instruction. How do you compare your generated answers with golden answers? Do you apply exact match or other methods?

**Limitations:**

The paper adequately describes its limitations.

---

> ### Author Rebuttal · Authors · 2024-08-06
>
> Many thanks for reviewing our work and providing these valuable feedback and comments.
>
> **W1**: In our method, information extraction is implemented using closed-source or publicly available LLMs with frozen parameters, significantly reducing computational expense.
> The primary resource-intensive component of our method is the BERT-based pruning model. However, it's important to highlight that this model contains only 110M parameters. This makes our approach less demanding in terms of computational resources. The frozen parameters LLMs-based information extraction and BERT-based pruning allow our method for real-time applications.
>
> **W2**: Due to the paper space limitation, we describe the primary details of the experiments in Section 4 (Experimental), including the statistical information of the datasets, the methodologies of the baseline approaches, the evaluation metrics, the detailed parameters, and the environment setting of the model training process. To enhance the clarity and understanding of our experiments, we also provide the code in the supplementary materials. In addition to these main experimental details, the other details of our experiments include: we use the Adam optimizer, and set the hidden size of the linear layer to 2048. In the revised version, we will include these experimental details in the appendix.
>
> **Q1**: The primary difference between our extraction and extraction into knowledge graph triples is that our extraction results are more centralized, whereas the knowledge graph triples are more scattered. The centralized extraction results of our method have two advantages. Firstly, our results aggregate content related to the same topic together. For example, as shown in Figure 2, all information related to the movie "Julius Caesar" is concentrated under the "Julius Caesar" node. Secondly, our extraction results include multi-hop connections, such as the path Julius Caesar → Producer→ Education in Figure 2, which is very helpful for answering complex the multi-hop question: Where did the producer of Julius Caesar study or work?. In the revised version, we will include a description of the differences between our method and the knowledge graph triples extraction method  (Leveraging Structured Information ... (Li et al., EMNLP 2023)) in the related work.
>
> **Q2**: The reported results in the paper are the means of the three times of each experiment. Based on the three times results of each experiment, the specific standard deviations for all datasets and LLMs involved in the paper are as follows:
>
> | LLMs | Methods | HOVER-2 | HOVER-3 | HOVER-4 | FEVEROUS | SCIFACT |
> |----------|----------|----------|----------|----------|----------|----------|
> | LLAMA2-70B    | InfoRE     | 52.83 &plusmn; 0.6     | 51.42 &plusmn; 1.64     | 50.04 &plusmn; 0.37     | 67.84 &plusmn; 0.16     | 63.81 &plusmn; 0.27     |
> | GPT-3.5    | InfoRE     | 68.21 &plusmn; 1.89     | 66.45 &plusmn; 1.53     | 64.91 &plusmn; 1.82     | 91.31 &plusmn; 0.57     | 81.54 &plusmn; 0.4     |
> | GPT-4    | InfoRE     | 75.87 &plusmn; 0.66     | 74.06 &plusmn; 1.62     | 73.08 &plusmn; 1.71     | 95.62 &plusmn; 0.25     | 93.67 &plusmn; 1.99     |
>
>
> | LLMs | Methods | 2WMHQA | MuSiQue | SQA | HQA | WIKIHOP |
> |----------|----------|----------|----------|----------|----------|----------|
> | LLAMA2-70B    | InfoRE     | 57.62 &plusmn; 0.52     | 52.78 &plusmn; 1.11     | 55.32 &plusmn; 1.77     | 69.98 &plusmn; 0.55     | 42.90 &plusmn; 0.12     |
> | GPT-3.5    | InfoRE     | 64.58 &plusmn; 1.62     | 58.03 &plusmn; 1.91     | 63.16 &plusmn; 1.31     | 77.12 &plusmn; 0.89     | 51.87 &plusmn; 0.32     |
> | GPT-4    | InfoRE     | 76.52 &plusmn; 1.1     | 66.36 &plusmn; 1.34     | 71.20 &plusmn; 1.44     | 83.22 &plusmn; 0.64     | 58.01 &plusmn; 0.93     |
>
> **Q3**: We don't limit the types of entities and relationships during extraction and apply open relationships. This is because closed relations restrict the relationships between entities to specific categories, making it challenging to express complex textual content. Limiting entity types further exacerbates this issue. We use a large language model based on the prompts in Appendix A for extraction. This extraction method was adopted after extensive analysis and experimentation. The extraction results aggregate content related to the same topic together and include multi-hop connections.
>
> **Q4**: In our experiments, we don’t separately assess completeness but instead focus on consistency. Specifically, we use GPT-4-32K to determine whether the extraction results are consistent with the original documents. If the results are found to be inconsistent, we re-execute the extraction process until consistent results are obtained. According to our statistics, we found that over 98% of the extraction results are consistent with the original text after the first extraction. This indicates that inconsistencies are rare. Due to the high consistency rate, we initially did not include this content in the paper. To improve the clarity of our method, we will include these details in the revised version.
>
> **Q5**: In our experiments, we use special tags <answer>{final_answer}</answer> to distinguish the final reasoning answer from intermediate results, as shown in Table 7. After using regular expressions to extract the final answer between the <answer></answer> tags, we combine it with the gold labels provided in the dataset and input them into the evaluation script to obtain the evaluation results. For the datasets 2WikiMultiHopQA, StrategyQA, HotpotQA, MuSiQue, and WIKIHOP, we run the official evaluation scripts provided by these datasets to get the evaluation results, including EM (exact match), F1, Precision, and Recall. For the HOVER, FEVEROUS, and SCIFACT datasets, we extract True or False predictions from final answer and use run classification evaluation scripts to get the evaluation results, including F1, Precision, and Recall.

---

> > ### Comment · Reviewer_wQkW · 2024-08-12
> > **Reply to Rebuttal**
> >
> > Thank you for the response. It has solved my primary concern. I have updated my rating score.

---

> > > ### Author Response · Authors · 2024-08-12
> > > **Appreciation for Positive Feedback**
> > >
> > > Dear Reviewer wQkW,
> > >
> > > Thank you for your positive reassessment, and we appreciate your acknowledgment of our work. We are truly grateful for your insightful feedback throughout the entire review process, which has helped us further improve our work.
> > >
> > > Best regards,
> > >
> > > The authors of InfoRE

---

### Author Rebuttal · Authors · 2024-08-07

Thank you very much to the reviewers for reviewing our work and providing valuable feedback and suggestions.

In the PDF file, we supplement the results of our method on the GPT-35-Turbo-0613 version.

---

### Decision · Program_Chairs · 2024-09-25

**Decision:**

Accept (poster)

**Comment:**

After carefully reading the paper and the author responses, all reviewers agree that this paper presents a novel method for re-organizing information before reasoning. The proposed algorithm demonstrated an average improvement of 4% in reasoning performance across 10 datasets and 3 large language models. It is independently effective and compatible with other reasoning strategies like CoT, offering a comprehensive breakdown of its components and their contributions.

There are still some rooms to further improve the paper. For example, it would be great if an in-depth comparison with knowledge-graph-based algorithms (e.g., GraphRAG (https://microsoft.github.io/graphrag/) can be included in the camera ready.